# Propagation of SARS-CoV-2 in Calu-3 Cells to Eliminate Mutations in the Furin Cleavage Site of Spike

**DOI:** 10.3390/v13122434

**Published:** 2021-12-04

**Authors:** John James Baczenas, Hanne Andersen, Sujatha Rashid, David Yarmosh, Nikhita Puthuveetil, Michael Parker, Rebecca Bradford, Clint Florence, Kimberly J. Stemple, Mark G. Lewis, Shelby L. O’Connor

**Affiliations:** 1Wisconsin National Primate Center, UW-Madison, Madison, WI 53711, USA; baczenas@wisc.edu; 2BIOQUAL, Inc., Rockville, MD 20852, USA; handersen@bioqual.com (H.A.); mlewis@bioqual.com (M.G.L.); 3American Type Culture Collection (ATCC), 10801 University Boulevard, Manassas, VA 20110, USA; srashid@atcc.org (S.R.); dyarmosh@atcc.org (D.Y.); nputhuveetil@atcc.org (N.P.); mparker@atcc.org (M.P.); rbradford@atcc.org (R.B.); 4The Biodefense and Emerging Infections Research Resources Repository (BEI Resources), 10801 University Boulevard, Manassas, VA 20110, USA; 5Office of Biodefense, Research Resources and Translational Research, Division of Microbiology and Infectious Diseases, NIAID, NIH, Bethesda, MD 20892, USA; clint.florence@nih.gov (C.F.); kstemple@niaid.nih.gov (K.J.S.); 6Department of Pathology and Laboratory Medicine, UW-Madison, Madison, WI 53705, USA

**Keywords:** SARS-CoV-2, polymorphisms, Calu-3, stock preparation

## Abstract

SARS-CoV-2 pathogenesis, vaccine, and therapeutic studies rely on the use of animals challenged with highly pathogenic virus stocks produced in cell cultures. Ideally, these virus stocks should be genetically and functionally similar to the original clinical isolate, retaining wild-type properties to be reliably used in animal model studies. It is well-established that SARS-CoV-2 isolates serially passaged on Vero cell lines accumulate mutations and deletions in the furin cleavage site; however, these can be eliminated when passaged on Calu-3 lung epithelial cell lines, as presented in this study. As numerous stocks of SARS-CoV-2 variants of concern are being grown in cell cultures with the intent for use in animal models, it is essential that propagation methods generate virus stocks that are pathogenic in vivo. Here, we found that the propagation of a B.1.351 SARS-CoV-2 stock on Calu-3 cells eliminated viruses that previously accumulated mutations in the furin cleavage site. Notably, there were alternative variants that accumulated at the same nucleotide positions in virus populations grown on Calu-3 cells at multiple independent facilities. When a Calu-3-derived B.1.351 virus stock was used to infect hamsters, the virus remained pathogenic and the Calu-3-specific variants persisted in the population. These results suggest that Calu-3-derived virus stocks are pathogenic but care should still be taken to evaluate virus stocks for newly arising mutations during propagation.

## 1. Introduction

The rapid development of vaccines and therapeutic measures to contend with the global SARS-CoV-2 pandemic is unprecedented. SARS-CoV-2 continues to evolve and adapt to the human population, such that there will be a need to assess the continued effectiveness of current medical countermeasures or improve them in the short and long term. This has been highlighted by the efforts to evaluate whether the first-generation SARS-CoV-2 vaccines are effective against the highly transmissible Delta variant [1,2].

Advancing new interventions relies on performing preclinical efficacy studies in both small and large animals [3,4]. Each animal study requires the use of a well-characterized challenge stock produced in cell culture, ideally with minimal passaging to minimize the rise of genetic sub-populations that may affect the phenotype of the virus stock. SARS-CoV-2 challenge stocks are propagated from an original virus isolate prepared from a clinical sample in a cell culture as a Passage 1 stock. This serves as a seed stock that is propagated to make larger challenge stocks. Many quality control analyses of these stocks are limited to sequencing either a portion of the genome or simply generating a consensus sequence. This is insufficient. Each stock of virus should be deep sequenced across the whole genome to identify variant subpopulations that can accumulate as the virus is passaged. The full sequence of the original isolate, the number of passages of virus, and the cell lines used for virus propagation can all influence the pathogenicity of the challenge stock that is produced. Comparing the population sequence of each virus stock against the original isolate is one of the most straightforward quality control measures that can be taken. This information can affect the downstream use of the stocks, as evidenced in the current study.

All early SARS-CoV-2 strains were initially propagated on versions of African Green Monkey Vero cell lines. Vero cells grow quickly and express high levels of ACE2, the molecule recognized by the RBD (receptor-binding domain) of the spike protein that SARS-CoV-2 uses to enter lung epithelial cells [5,6,7]. Viral entry via the ACE2 receptor is improved by the furin-mediated pre-activation of the spike protein [8]. However, SARS-CoV-2 can also enter Vero cells by endocytosis, rendering the need to use the ACE2 receptor irrelevant [9,10]. Several groups have found that the propagation of SARS-CoV-2 on Vero-derived cells lines leads to the rapid accumulation and loss of a functional furin cleavage site within a few passages [11,12,13,14]. As a result, the propagation of SARS-CoV-2 on Vero cells gave rise to challenge stocks with defects in the furin cleavage site, making them non-pathogenic in animals [15].

To reduce the likelihood that SARS-CoV-2 challenge stocks acquire mutations in the furin cleavage site, some groups have grown SARS-CoV-2 on modified Vero cell lines. Vero/hSLAM cells were originally used to successfully grow a strain from Victoria, Australia that did not accumulate defective mutations in the furin cleavage site [16]. Even though Vero/hSLAM cells can support replication, a better Vero derivative was later found to be Vero/TMPRSS2 cells [17]. The TMPRSS2 serine protease primes the spike protein prior to the recognition of the ACE2 receptor and cell entry [8,9,18]. SARS-CoV-2 virus isolates passaged in cells expressing TMPRSS2 use this pathway for entry and retain the furin cleavage site [19]. In Ad-ACE2-transduced mice lacking TMPRSS2, SARS-CoV-2 growth is inhibited [20], implying that the TMPRSS2-mediated pathway is a key component of pathogenic infection in vivo.

Calu-3 cells are a human lung epithelial cell line that can support the propagation of SARS-CoV-2 and be a superior option for virus propagation. These cells can promote replication of SARS-CoV-2 viruses or pseudoviruses expressing the spike gene with an intact furin cleavage site, but viruses lacking this cleavage site do not efficiently replicate in Calu-3 cells [10,21,22]. While Calu-3 cells grow more slowly than Vero-derived cell lines, this deficiency is offset by the value of maintaining the furin cleavage site in the virus stocks.

In the present study, we evaluated how the propagation of a B.1.351-related (beta variant) SARS-CoV-2 isolate in two cell types, Vero/hSLAM and Calu-3 cells, affected viral genome sequences both within and outside of the furin cleavage site of the spike gene. We found an expansion of mutations in the furin cleavage site when the virus was grown on Vero/hSLAM cells, but these were eliminated when propagating it in Calu-3 cells. We also found distinct variants that accumulated when this virus stock was grown on Calu-3 cells. Importantly, the Calu-3-derived virus stocks remained pathogenic in hamsters, and the Calu-3-specific variants were maintained. Together, these results support the growth of SARS-CoV-2 challenge stocks on Calu-3 cells prior to use in animals to retain an intact furin cleavage site and viral pathogenicity.

## 2. Materials and Methods

### 2.1. Reference Sequence for hCoV-19/South Africa/KRISP-K005325/2020 Used in This Study

The complete genome of the clinical isolate of SARS-CoV-2, hCoV-19/South Africa/KRISP-K005325/2020 has been sequenced (GISAID: EPI_ISL_678615). The following mutations are reported to be present in the clinical isolate: Spike A243del, Spike A701V, Spike D80A, Spike D215G, Spike D614G, Spike E484K, Spike K417N, Spike L18F, Spike L242del, Spike L244del, Spike N501Y, E (envelope protein) P71L, N (nucleocapsid protein) T205I, NSP3 (non-structural protein 3) Q57H, NSP3 S171L, NSP3 W131L, NSP7a (non-structural protein 7a) V93F, NSP2 (non-structural protein 2) T85I, NSP3 K837N, NSP5 (non-structural protein 5) K90R, NSP6 (non-structural protein 6) F108del, NSP6 G107del, NSP6 S106del, and NSP12 (non-structural protein 12) P323L. The deposited virus (at passage three) was reported to have additional mutations compared to the clinical isolate: N R32H, ORF1a (open reading frame 1a) N4358K, and ORF9b (open reading frame 9b) A29T.

### 2.2. Propagation of hCoV-19/South Africa/KRISP-K005325/2020 by BEI Resources

Severe acute respiratory syndrome-related coronavirus 2 (SARS-CoV-2), isolate hCoV-19/South Africa/KRISP-K005325/2020 (also referred to as 501Y.V2.HV and 501Y.V2.HV001) was isolated from an oropharyngeal swab from a 40-year-old human male in Ugu district, KwaZulu-Natal, South Africa on 16 November 2020. Under the nomenclature system introduced by GISAID (Global Initiative on Sharing All Influenza Data), SARS-CoV-2, isolate hCoV-19/South Africa/KRISP-K005325/2020 was assigned lineage B.1.351 and GISAID clade GH using Phylogenetic Assignment of Named Global Outbreak LINeages (PANGOLIN) tool. The viral strain was isolated and passaged three times at the Africa Health Research Institute, Durban 4001, South Africa [23]. H1299-ACE2-E3 cells were used for initial isolation followed by two passages into Vero E6 cells (ATCC^®^ CRL-1586™) prior to its deposition with BEI Resources (Manassas, VA, USA), a US-funded biorepository for production and distribution to the scientific community.

Using the virus deposited with BEI Resources (hereinafter referred to as “hCoV-19/South Africa/KRISP-K005325/2020 p4”), we inoculated both Calu-3 (ATCC^®^ HTB-55™) and Vero-hSLAM cells, which are *Cercopithecus aethiops* kidney epithelial cells with human signaling lymphocytic activation molecule (SLAM), also known as CDw150. Vero/hSLAM cells were generously donated to BEI Resources by the CDC and are available in the European Collection of Authenticated Cell Cultures (ECACC Cat #04091501).

The viral growth in Calu-3 cells had a multiplicity of infection (MOI) of 0.00001 based on the titer of the Passage 3 virus received from the depositing laboratory that had isolated and made virus stocks. Once the virus inoculum was adsorbed for one hour at 37 °C, viral growth media consisting of Eagle’s Minimum Essential Medium (ATCC^®^ 30-2003™) supplemented with 2% fetal bovine serum (ATCC^®^ 30-2020™) were added. After 5 days of incubation at 37 °C with 5% CO_2_, when roughly 60% of the cell monolayer showed cytopathic effects, the cell lysate and supernatant were harvested. Cell lysate and supernatant was clarified by centrifuging at 1500× *g* for 10 min at room temperature.

The viral growth in Vero-hSLAM cells had a multiplicity of infection (MOI) of 0.00001, adsorbed for one hour at 37 °C before adding viral growth media consisting of Eagle’s Minimum Essential Medium (ATCC^®^ 30-2003™) supplemented with 2% fetal bovine serum (ATCC^®^ 30-2020™). After 4 days of incubation at 37 °C with 5% CO_2_, when roughly 75% of the cell monolayer showed cytopathic effects, the cell lysate and supernatant were harvested and spin-clarified by centrifugation at 1500× *g* for 10 min at room temperature, and the resulting virus stock rapidly frozen using liquid nitrogen in small aliquots.

These viral strains (representing Passage 4) are available from BEI Resources under the catalog numbers NR-54009 (Vero-hSLAM) and NR-54974 (Calu-3). Genomic RNA (gRNA) was extracted for sequencing from a preparation of cell lysate and supernatant from each propagation using QIAamp^®^ Viral RNA Mini Kit (QIAGEN 52906).

### 2.3. Sequencing and Bioinformatic Analysis of hCoV-19/South Africa/KRISP-K005325/2020 Stocks Propagated by BEI Resources

The ATCC Sequencing and Bioinformatics Center (SBC) performed the library prep, sequencing, and bioinformatics analysis. SARS-CoV-2 samples were prepared for sequencing using the NEBNext^®^ Ultra II™ RNA Library Prep Kit for Illumina^®^ (NEB #E7775) according to the manufacturer’s protocols. The samples were sequenced on the Illumina MiSeq platform using either a MiSeq^®^ Reagent Micro Kit v2 (300 Cycle) or a MiSeq^®^ Reagent Kit v2 (500 cycle) run for 300 cycles. Produced reads were trimmed using a custom pipeline, comprising Trimmomatic v0.38 using default parameters and FastQC v0.11.9, to remove low-quality bases and adapter sequences. Reference-based assembly was performed using the following genomes as reference sequences: MN908947.3 (severe acute respiratory syndrome coronavirus 2 isolate Wuhan-Hu-1, complete genome) from GenBank and the clinical reference sequence for the interrogated strain.

Reads were then mapped to each reference sequence using bwa mem v0.7.17-r1188, one of the short-read aligner algorithms in the Burrow–Wheeler Aligner (BWA) software package designed for mapping reads or sequences against a large reference genome. Bwa mem was run with paired-end reads and with default parameters. Though the default alignment parameters are very relaxed and nonspecific, BEI Resources believes this to be both acceptable and appropriate for aligning reads from an isolate source prior to assembly. Reference regions with <10× mean coverage were considered poorly supported low coverage regions and excluded from further analysis. Reference coverage statistics were reported for each set of mapped reads using Qualimap v.2.2.1, a Java and R application, which provided sequence alignment quality metrics.

LoFreq Viterbi was then used to perform a series of global alignment-level realignments that considered the context of each read alignment to improve mismatch or indel positional agreement. Once completed, further analysis was performed by identifying variants relative to the respective reference sequence. In order to maintain consistent reporting of nucleotide location for the purposes of comparison and analysis, all variants were globally reported in the standard format, i.e., relative to the sequence location within the SARS-CoV-2 Wuhan-Hu-1 reference genome (MN908947.3). For a nucleotide at position X, the variant frequency was calculated as: Variant Frequency = (# Reads with Variant Nucleotide Mapped to Position X)/(Total # Reads Mapped to Position X).

SNPs and indels were detected using LoFreq version 2.1.8, an ultra-sensitive variant caller that makes use of base, indel, mapping, and alignment qualities to aid in the detection of low-frequency variants. The SBC variant calling pipeline incorporated LoFreq according to its suggested best practices. The pipeline first calls bcftools mpileup to generate an initial variant call set. Bcftools mpileup is one of the commands in the bcftools utilities set (version 1.10.2) that is designed to identify all nucleotides from a BAM file by position along a reference sequence. The command is used to only output sites with variant alleles present, which are then supplied to the Genome Analysis Toolkit Base Quality Score Recalibration (GATK BQSR) tool, the next step of the pipeline. Briefly, the GATK BQSR tool applies machine learning techniques to correct and adjust quality scores that can often be inaccurate due to sequencing errors. Once quality scores are adjusted, LoFreq calls variants using the realigned reads and can then identify lower-frequency variants, though it must be noted that variants below 5% frequency are still considered spurious.

In certain cases, some samples contained indels that LoFreq could not identify. FreeBayes v1.3.1-dirty was used to validate LoFreq variant calls and capture these indels. FreeBayes is a Bayesian genetic variant detector designed to find small polymorphisms, specifically SNPs (single-nucleotide polymorphisms), indels (insertions and deletions), MNPs (multi-nucleotide polymorphisms), and complex events (composite insertion and substitution events) smaller than the length of a short-read sequencing alignment. Since LoFreq is particularly sensitive to low-frequency variations and FreeBayes is adept at capturing indels, the final variants list is a merger of the results from both variant callers. We ran BEI Resources and collaborator read sets through this pipeline and found substantial agreement between BEI Resources detected variants and collaborator variant calls.

### 2.4. Propagation of hCoV-19/South Africa/KRISP-K005325/2020 by BIOQUAL

Virus stocks were generated by infecting Calu-3 cells (ATCC HTB-55) in an EMEM medium containing 2% FBS, L-glutamine, and penicillin/streptomycin. Specifically, 10 mL of diluted seed stock were added to each T150 flask with ~80% confluent cell monolayer and allowed to absorb for 1 h at 37 °C and 5% CO_2_. An additional 42 mL of EMEM with 2% FBS were added to each flask, and the cell cultures were allowed to incubate for 3–4 days depending upon the amount of the nucleocapsid (NP) protein present in the supernatant, as measured by the antigen capture kit (My BioSource). The culture medium was collected into 50 mL conical tubes and centrifuged at 1500 rpm for 10 min at 4 °C. The supernatants were combined into a large flask and mixed well, and 0.5–1 mL aliquots were prepared in 2 mL cryovials. The resulting virus stock was tested for sterility and stored at −80 °C.

### 2.5. Infection of Syrian Hamsters with BQ-RSA-p4

Four Golden Syrian hamsters (2 male and 2 female) were intranasally infected with 0.1 mL of a 1:10 dilution of the BQ-RSA-p4 stock. The titer of the neat stock was 5 × 10^8^ TCID50/mL in Vero-TMPRSS2 cells, which was 1.09 × 10^7^ PFU/mL in Vero-TMPRSS2 cells, which means that 5 × 10^6^ TCID50 of virus was used to infect each animal. Animals were observed and weighed daily. Oral swabs were collected on days 2, 4, and 7 to measure viral loads using quantitative PCR for genomic and subgenomic RNA, as described previously [24]. At 7 days post infection, animals were euthanized and maximally bled. Lungs were collected at necropsy. Viral RNA was isolated from lung for sequencing and viral load testing.

### 2.6. Sequencing and Bioinformatic Analysis of hCoV-19/South Africa/KRISP-K005325/2020 Stocks Propagated by BIOQUAL

SARS-CoV-2 inocula were sequenced using a modified approach originally developed by ARTIC Network (https://artic.network, accessed on 12 February 2021) [14]. Briefly, complementary DNA (cDNA) was synthesized with SuperScript IV Reverse Transcriptase (Invitrogen, Carlsbad, CA, USA), random hexamers, and dNTPs. Complementary DNA was PCR-amplified with a multiplex PCR amplicon-based approach that was developed for Nanopore. The multiplex PCR used a total of 96 primers in two pools with Q5 Hot Start Hi-Fi 2x Master mix (New England Biolabs, Ipswich, MA, USA) and the following thermocycling conditions: 98 °C for 30 s, followed by 25 cycles of 98 °C for 15 s and 65 °C for five minutes, then an indefinite hold at 4 °C. Amplified PCR products were pooled and purified with AMPure XP beads (Beckman Coulter, Brea, CA, USA). Libraries were prepared with the TruSeq sample preparation kit (Illumina, San Diego, CA, USA). Samples were end-repaired, purified using the Sample Purification Beads (SPB), and A-Tailed followed by adaptor ligation. A post-ligation bead cleanup was performed with SPBs. Finally, each sample was amplified via eight cycles of PCR, cleaned with SPBs, and eluted in RSB. The concentration and average fragment length were determined with a Qubit dsDNA high-sensitivity kit (Invitrogen, Waltham, MA, USA) and Agilent’s High Sensitivity DNA kit, respectively. Each sample was equimolarly pooled to a concentration of 4 nM. This pool was denatured with five μL of 0.2 N NaOH, vortexed, and incubated at room temperature for five minutes. An HT1 buffer solution was added to generate a 20 pM pool. The 20 pM pool was then diluted to a final concentration of 10 pM, and a Phix-derived control was spiked in to account for 10% of the total DNA. The pool was loaded onto a 2 × 250 cycle V2 cartridge to be sequenced on an Illumina MiSeq.

An analytical pipeline called “Zequencer_ncov19” was developed to process the raw FASTQ files. In short, the primer sequences were trimmed and the reads were paired and merged using BBDuk (https://jgi.doe.gov/data-and-tools/bbtools/bb-tools-user-guide/bbduk-guide/ (accessed on 29 October 2021)) and BBMerge (https://jgi.doe.gov/data-and-tools/bbtools/bb-tools-user-guide/bbmerge-guide/, accessed on 29 October 2021), respectively. The reads were then mapped to the Wuhan-1 reference genome (MN908947.3) using BBMap (https://jgi.doe.gov/data-and-tools/bbtools/bb-tools-user-guide/bbmap-guide/ (accessed on 29 October 2021)). Variants and SNPs were detected with SnpEff and VarScan, and they were called at a threshold of 1%. The BAM alignments and vcf files were imported into Geneious Prime 2021.1.1, which was used to visualize and compare the alignments to the SARS-Related Coronavirus 2 isolate (hCoV-19/South Africa/KRISP-K005325/2020) reference genome, obtained from GISAID. A version of the Zequencer_ncov19 analysis pipeline is available at https://github.com/DABAKER165/zequencer_ncov19 (accessed on 29 October 2021). Sequences were uploaded to SRA, as indicated in Appendix A and metrics are shown in Appendix A.

## 3. Results

### 3.1. hCoV-19/South Africa/KRISP-K005325/2020 Passage Description

The workflow for passaging the hCoV-19/South Africa/KRISP-K005325/2020 (EPI_ISL_678615) stocks is shown in Figure 1. As described in the Methods section, this virus stock was isolated from an individual in South Africa and then propagated once in H1299-ACE2-E3 cells and twice more in Vero E6 cells. This Passage 3 (p3) virus stock was independently received by both BEI Resources and BIOQUAL.

### 3.2. hCoV-19/South Africa/KRISP-K005325/2020, Propagation, and Sequencing at BEI Resources

At BEI Resources, the deposited p3 stock was propagated separately on Vero-SLAM (Catalog #: NR-54009) and Calu-3 (Catalog #: NR-54974) cells to generate two independent p4 stocks. A sample of virus from each p4 stock was deep sequenced, as described in the Methods section. Comparative analyses of the variants present in both p4 stocks were performed to two different reference sequences: (1) the Wuhan-Hu-1 reference sequence (Genbank: MN908947) and (2) the hCoV-19/South Africa/KRISP-K005325/2020 reference reported in GISAID (EPI_ISL_678615).

The following fixed mutations (Table 1) were confirmed to be present, as reported in the clinical sample in both the Vero-hSLAM (NR-54009)- and Calu-3 (NR-54974)-derived p4 stocks propagated by BEI Resources: Spike A243del, Spike A701V, Spike D80A, Spike D215G, Spike D614G, Spike E484K, Spike K417N, Spike L18F, Spike L242del, Spike L244del, Spike N501Y, E (envelope protein) P71L, N (nucleocapsid protein) T205I, ORF3 (open reading frame 3) Q57H, NSP3 S171L, NSP3 W131L, ORF7a (open reading frame 7a) V93F, NSP2 (non-structural protein 2) T85I, NSP3 K837N, NSP5 (non-structural protein 5) K90R, NSP6 (non-structural protein 6) F108del, NSP6 G107del, NSP6 S106del, and NSP12 (non-structural protein 12) P323L. The deposited virus (at passage 3) was reported to have additional mutations compared to the clinical isolate: N R32H, ORF1a (open reading frame 1a) N4358K, and ORF9b (open reading frame 9b) A29T.

There were additional variants detected at a frequency of greater than 5% in the NR-54009, NR-54974, or both p4 stocks that were not described in the clinical reference consensus sequence (Table 2, Table 3 and Table 4). Importantly, there were two mutations at positions 23,596 and 23,606, which are located just upstream and within the furin cleavage site. Variants were present in the p4 stock grown on Vero-hSLAM cells (NR-54009) at approximately 90% (Table 2), but they were absent from the Calu-3-derived NR-54974 stock (Table 3). Table 4 shows variants present in both stocks, but the variant frequencies were not necessarily identical between stocks.

### 3.3. hCoV-19/South Africa/KRISP-K005325/2020 Propagation and Sequencing at BIOQUAL

BIOQUAL received the same p3 stock obtained at BEI Resources and serially propagated it twice on Calu-3 cells to generate Passages 4 and 5 (BQ-RSA-p4 and BQ-RSA-p5), which were sequenced by Illumina (see the Methods section). We analyzed the sequences relative to the Wuhan-Hu-1 reference sequence, as described in the Methods section. We found the same 25 variants associated with this virus lineage in both the BQ-RSA-p4 and -p5 stocks that were present in the NR-54009 and NR-54974 stocks produced at BEI Resources (Table 1 and Table 5).

In BQ-RSA-p4 and -p5, we detected the same four variants that BEI Resources found at a frequency of 90% or greater in the NR-54974 stock that had been propagated on Calu-3 cells (Table 4 and Table 6, positions 11,020, 13,339, 28,368, 29,821), but were at a frequency of 26% or less in the NR-54009 stock prepared on Vero-SLAMs and were absent from the Wuhan-Hu-1 or the EPI_ISL_678615 reference sequences. This suggests that these four variants may have arisen when growing the hCoV-19/South Africa/KRISP-K005325/2020 stock on Calu-3 cells at two different locations (BEI Resources and BIOQUAL) and raises the possibility that these variants reflect the adaptation of the B.1.351 virus lineage to Calu-3 cells. Importantly, the two mutations in the furin cleavage site that were detected at 90% in NR-54009 grown on Vero-hSLAM cells (Table 2, positions 23,593 and 23,606) were only present at ~3% in BQ-RSA-p4 and undetectable in BQ-RSA-p5. This further supports evidence that SARS-CoV-2 viruses with variants in the furin cleavage site are unfit to grow on Calu-3 cells [22].

We also detected eight de novo variants that were in BQ-RSA-p5 at a frequency of greater than 5% but were absent in the Wuhan-Hu-1 or the EPI_ISL_678615 reference sequences (Table 7). When we examined the frequency of these same eight variants in BQ-RSA-p4, they were present at a lower frequency, except for site 26,250 (Table 7). These same eight variants were not detected in NR-54009 or NR-54974 prepared at BEI. Further studies will need to determine if these variants also represent adaptations to Calu-3 cells or if they arose by another mechanism.

BIOQUAL also received the NR-54974 stock from BEI Resources and propagated it once on Calu-3 cells to generate NR-54974-p5. This new stock was also sequenced and analyzed using the same methods as BQ-RSA-p4 and -p5. We found the same 25 fixed variants and 4 Calu-3-specific variants shown in Table 5 and Table 6, respectively. Of the eight de novo variants (Table 7), the synonymous variant at site 26,250 was present at 11% in BQ-RSA-p5 and at 45% in NR-54974-p5. The other seven de novo variants in BQ-RSA-p5 were not present in NR-54974-p5. Instead, NR-54974-p5 had six unique de novo variants present at a frequency of 5% or greater (Table 8). Homopolymeric regions were excluded.

### 3.4. BQ-RSA-p4 Retained Pathogenicity in Hamsters

Four hamsters (two males and two females) were intranasally infected with a 1:10 dilution of BQ-RSA-p4. Animals were monitored for 7 days and then euthanized. During the course of infection, we found oral viral titers were greater than 1 × 10^6^ copies per swab. At necropsy, we detected approximately 2 × 10^9^ copies of virus per gram of lung tissues, consistent with other studies of the SARS-CoV-2 infection of hamsters (Figure 2) [15,25,26,27]. Additionally, infected hamsters lost 15% of their body weight during the study, as expected for a pathogenic SARS-CoV-2 that has an intact furin cleavage site [14,15].

At necropsy, lungs were collected, and RNA was isolated for sequencing. Sequencing was performed by Illumina using the ARTIC protocol. Unfortunately, because these were samples isolated from tissues from later stages of infection, amplification was variable across the genome, and some amplicons dropped out completely. Sequence data can be found in the Bioproject PRJNA758017, and basic metrics are found in Appendix A.

We first examined the 25 fixed variants described in Table 5. At each of the 25 sites where we could confidently call variants (see the Methods section), the variants remained fixed in the viruses isolated from the animals (Appendix A).

We then examined the four nucleotide sites described in Table 6 that were specific to the virus stocks grown in Calu-3 cells. Coverage for the one site in the 3′UTR (position 29,821) was less than 70x for all datasets, but the variant was largely present. Of the other three sites, all were present at frequencies similar to the stock in viruses isolated from the hamster lungs (Appendix A).

Finally, the eight de novo variants described in Table 7 were examined. There were three sites with little to no coverage (2969, 21,633, and 23,044). Of the other five sites, four of them expanded in the animals, providing additional evidence that there BQ-RSA-p4 replicated in the hamsters. Only one site, 26,250, remained at a fairly stable frequency (Appendix A). Interestingly, this was the same site where the variant frequency did not increase when BQ-RSA-p4 was passaged once on Calu-3 cells to make BQ-RSA-p5, but it did increase when NR-54974 was passaged once on Calu-3 cells at BIOQUAL to produce NR-54974-p5. This suggests that the overall passage history can affect the final frequency of a variant in the virus population.

## 4. Discussion

The rapid acceleration of SARS-CoV-2 studies in animals to understand pathogenesis and develop therapeutics and vaccines has been essential to the pandemic response since early 2020. In the early days of the outbreak, the immediate need to study SARS-CoV-2 required that this virus be propagated as quickly as possible. As a result, Vero cells were the initial mainstay for propagating SARS-CoV-2 because these cells had been previously used for the propagation of closely related human coronaviruses [28]. At the time, the importance of an intact furin cleavage site for SARS-CoV-2 pathogenicity was underappreciated, and the extensive array of mutations and deletions in the furin cleavage site during the propagation of SARS-CoV-2 on various Vero-derived cells was unpredictable, likely due to the heterogeneity of the cells themselves in stocks available in laboratories globally. It has now become clear that the maintenance of an intact furin cleavage site is needed for SARS-CoV-2 to infect and replicate in animal models [15,29].

Appropriate virus propagation conditions are needed to maintain an intact furin cleavage site in SARS-CoV-2 virus populations. In this study, we evaluated whether the propagation of a B.1.351 lineage of SARS-CoV-2 on Calu-3 cells would yield a virus stock devoid of furin cleavage site mutations and deletions, even when the stock was previously passaged twice on Vero cells. Specifically, we wanted to determine whether the propagation on Calu-3 cells would eliminate accumulated virus subpopulations that have mutant furin cleave site sequences and prevent the generation of viruses with new variants or deletions of this site as a result of cell culture adaptations.

Two separate laboratories (BEI Resources and BIOQUAL) obtained the same p3 isolate of a B.1.351 virus lineage that had been passaged twice through Vero E6 cells. When independently grown on Calu-3 cells at each site, previously accumulated mutations in the furin cleavage site were lost from the population. However, BEI Resources also grew this same stock on Vero-hSLAM cells, only to find that the mutations in the furin cleavage site persisted in the population. Our results are consistent with recent studies that grew an early SARS-CoV-2 isolate and engineered pseudovirus mutants on Calu-3 cells. In each of these studies, SARS-CoV-2 viruses with an intact furin cleavage site infected Calu-3 cells faster than those viruses lacking these features [21,22]. We built on this previous observation by finding that B.1.351 ‘Beta’ SARS-CoV-2 variants with furin cleavage site point mutations were also lost from the population when propagated on Calu-3 cells. This observation indicates that likely all SARS-CoV-2 variants of concern will accumulate furin cleavage site mutations when grown on Vero cells, but these can be eliminated if propagated on Calu-3 prior to use in animals.

One concern about growing SARS-CoV-2 on Calu-3 cells is that other variants may arise in the virus population that reflect Calu-3 cell culture adaptations. We found four new variants present in all hCoV-19/South Africa/KRISP-K005325/2020-derived stocks grown on Calu-3 cells, including those grown at BEI Resources and BIOQUAL (Table 6). We hypothesize that these four variants are Calu-3 cell culture adaptations; they included two synonymous variants in Nsp6 and the 3′UTR and two nonsynonymous variants in Nsp10 and nucleocapsid. We also detected unique variants in the viruses propagated on Calu-3 cells in RSA-BQ-p4 and p5, but these were not present in the stocks propagated on Calu-3 cells at BEI Resources (Table 7).

To evaluate whether the Calu-3-specific mutations would persist in vivo, we infected four hamsters with the Calu-3-derived BQ-RSA-p4 stock. This virus grew well in these animals, with subgenomic viral titers exceeding 10^7^ copies/gram lung tissue—results similar to those found in hamsters infected with the WA/2020 (Wuhan) isolate [15,26,27]. These hamsters lost ~15% of their body weight, further suggesting that the Calu-3-associated mutations did not affect virus pathogenicity in hamsters. This stands in stark contrast to the limited weight loss observed in hamsters infected with stocks lacking a furin cleavage site [14,15]. The variants detected in the Calu-3-derived stocks were also either maintained or expanded in the hamsters. Future studies will need to evaluate whether other SARS-CoV-2 lineage variants (such as B.1.617.2) would acquire the same nucleotide variants when grown on Calu-3 cells or whether the nucleotide variants described here are restricted to the B.1.351 lineage.

Finally, our data further support the ongoing need to deep sequence virus challenge stocks prior to challenging animals. At one end of the spectrum, the loss of a functional furin cleavage site is an extreme but real consequence of growing SARS-CoV-2 stocks in Vero cells. Here, we found that growing SARS-CoV-2 in Calu-3 cells did not lead to the same furin cleavage site mutations, but there were other cell-specific adaptations. While these Calu-3-derived variants did not have apparent detrimental effects on pathogenesis in the hamsters studied here, that could not be predicted by simply looking at the sequence data.

Our study supports a workflow where the deep sequencing of a challenge inoculum to generate a comprehensive view of the complexity of the virus population is warranted. As SARS-CoV-2 challenge stocks are constantly propagated at different sites, variant frequencies will continue to change as stocks adapt to cell culture conditions. It is important to distinguish whether a pathogenic viral variant detected in an animal arose de novo or was originally present in the inoculum. As such, understanding virus diversity beyond the level of the consensus sequence is essential to identify underlying mutations that may alter virus pathogenicity. Thus, we propose that characterizing the complexity of the challenge stock should be required for every in vivo challenge study.

## Figures and Tables

**Figure 1 viruses-13-02434-f001:**
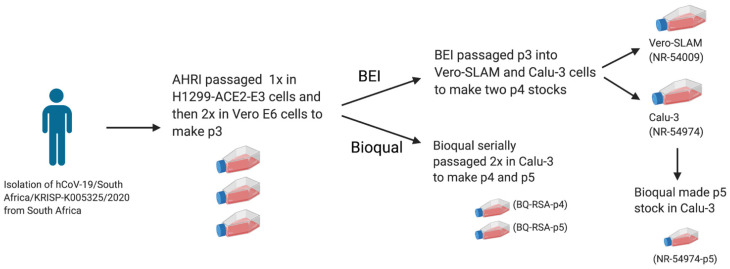
Schematic of the passage history for the virus stocks characterized in this study. Created with BioRender.com.

**Figure 2 viruses-13-02434-f002:**
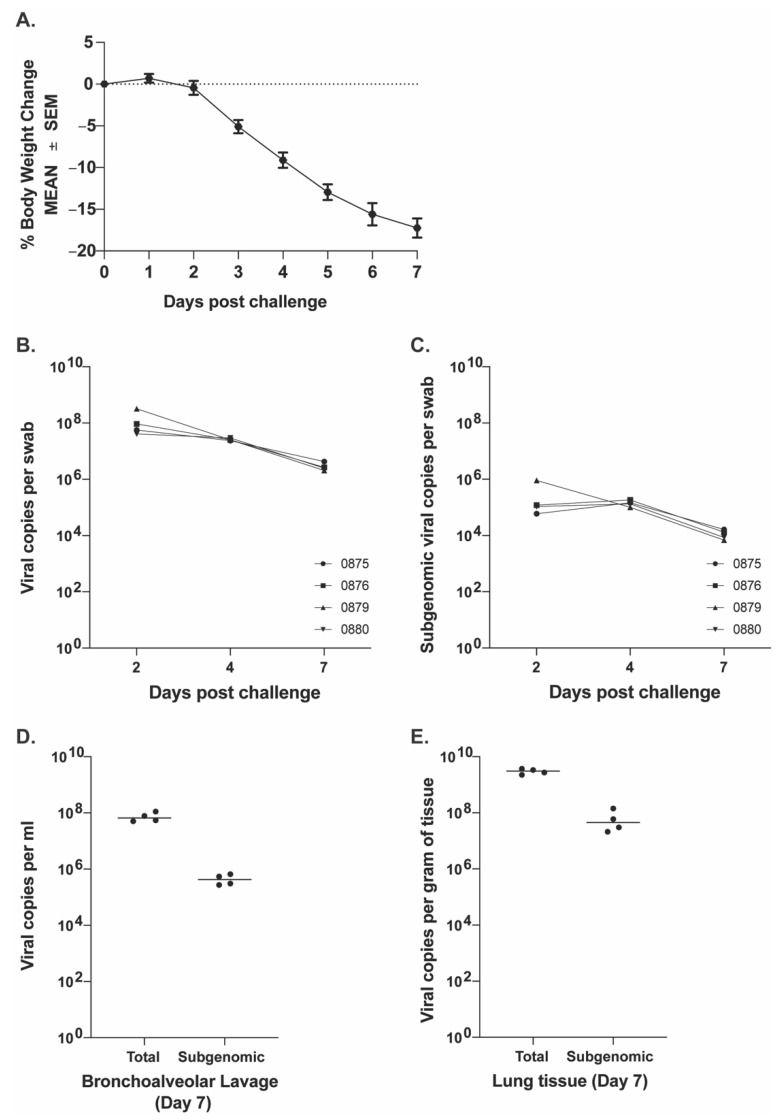
Hamsters are susceptible to pathogenic infection with BQ-RSA-p4 grown on Calu-3 cells. (**A**) The mean and SEM of the body weight for all four hamsters infected with BQ-RSA-p4 over the 7 days post challenge are shown. The dotted line represents zero change in body weight. Total (**B**) and subgenomic (**C**) viral copies from oral swabs were measured during the course of infection for all four hamsters listed. On the day of necropsy, the total and subgenomic viral copies were measured in the bronchoalveolar lavage fluid (**D**) and lung tissue (**E**) for all four hamsters. Each dot represents a different animal, and the median is shown with the line. Methods from Alleva et al. [24] were used for virus quantification.

**Table 1 viruses-13-02434-t001:** Twenty-five fixed Variants in both NR-54009 and NR-54974 consistent with the clinical reference sequence (EPI_ISL_678615). (* indicates deletion).

Position in MN908947 Wuhan-Hu-1 Sequence	Position in EPI_ISL_ 678615 Reference Sequence	Reported MN908947 Wuhan-Hu-1 Sequence	Reported EPI_ISL_ 678615 Reference Sequence	Nucleotide in NR-54009 and NR-54974	Gene	Amino Acid Mutation
1059	1057	C	T	T	Nsp2	T85I
5230	5228	G	T	T	Nsp3	K837N
10,323	10,321	A	G	G	Nsp5	K90R
14,408	14,397	C	T	T	Nsp12	P323L
21,614	21,603	C	T	T	Spike	L18F
21,801	21,790	A	C	C	Spike	D80A
22,206	22,195	A	G	G	Spike	D215G
22,813	22,793	G	T	T	Spike	K417N
23,012	22,992	G	A	A	Spike	E484K
23,063	23,043	A	T	T	Spike	N501Y
23,403	23,383	A	G	G	Spike	D614G
23,664	23,644	C	T	T	Spike	A701V
25,563	25,543	G	T	T	ORF3a	Q57H
25,784	25,764	G	T	T	ORF3a	W131L
25,904	25,884	C	T	T	ORF3a	S171L
26,456	26,436	C	T	T	E	P71L
27,670	27,650	G	T	T	ORF7a	V93F
28,887	28,867	C	T	T	N	T205I
11,287 *	11,285 *	GTCTGGTTTT	G	G	Nsp6	ΔSGF (aa106–108)
22,286 *	22,275 *	CTTGCTTTAC	C	C	Spike	ΔLAL (aa242–244)
174	172	G	T	T	5′UTR	No AA Change
241	239	C	T	T	5′UTR	No AA Change
2692	2690	A	T	T	Nsp2	No AA Change
3037	3035	C	T	T	Nsp3	No AA Change
28,253	28,233	C	T	T	ORF8	No AA Change

**Table 2 viruses-13-02434-t002:** Variants present in NR-54009 but not present in the clinical reference consensus sequence (EPI_ISL_678615).

Position in MN908947 Wuhan-Hu-1 Sequence	Position in EPI_ISL_ 678615 Reference Sequence	Reported MN908947 Wuhan-Hu-1 Sequence	Reported EPI_ISL_ 678615 Reference Sequence	Nucleotide in NR-54009	Variant Frequency	Gene	Amino Acid Mutation
10,809	10,807	C	C	T	54%	Nsp5	P252L
11,750	11,739	C	C	T	10%	Nsp6	L260F
17,339	17,328	C	C	T	7%	Nsp13	A368V
21,651	21,640	A	A	C	12%	Spike	N30T
23,593	23,573	G	G	T	90%	Spike	Q677H
23,606	23,586	C	C	T	90%	Spike	R682W
25,810	25,790	C	C	T	14%	ORF3a	L140F
26,822	26,802	C	C	T	7%	M	No AA Change
26,984	26,964	C	C	T	6%	M	No AA Change
27,393	27,373	C	C	T	63%	btw ORF6/7	No AA Change
27,627	27,607	T	T	A	28%	ORF7a	No AA Change

**Table 3 viruses-13-02434-t003:** Variants present in NR-54974 but not present in the clinical reference consensus sequence (EPI_ISL_678615).

Position in MN908947 Wuhan-Hu-1 Sequence	Position in EPI_ISL_ 678615 Reference Sequence	Reported MN908947 Wuhan-Hu-1 Sequence	Reported EPI_ISL_ 678615 Reference Sequence	Nucleotide in NR-54974	Variant Frequency	Gene	Amino Acid Mutation
3721	3719	T	T	C	9%	Nsp3	No AA Change
8821	8819	A	A	G	9%	Nsp4	No AA Change
10,082	10,080	T	T	C	5%	Nsp5	S10L
10,451	10,449	A	A	G	9%	Nsp5	N133D
15,909	15,898	T	T	C	6%	Nsp12	No AA Change

**Table 4 viruses-13-02434-t004:** Variants present in both NR-54009 and NR-54974 but not present in the clinical reference consensus sequence (EPI_ISL_678615).

Stock Number	Position in MN908947 Wuhan-Hu-1 Sequence	Position in EPI_ISL_ 678615 Reference Sequence	Reported MN908947 Wuhan-Hu-1 Sequence	Reported EPI_ISL_ 678615 Reference Sequence	Nucleotide in NR-54009 or NR-54974	Variant Frequency	Gene	Amino Acid Mutation
NR-54974	1963	1961	T	T	C	9%	Nsp2	No AA Change
NR-54974	1963	1961	T	T	G	6%	Nsp2	No AA Change
NR-54009	1963	1961	T	T	A	6%	Nsp2	No AA Change
NR-54974	11,020	11,018	C	C	T	95%	Nsp6	No AA Change
NR-54009	11,020	11,018	C	C	T	25%	Nsp6	No AA Change
NR-54974	13,339	13,328	T	T	G	94%	Nsp10	N105K
NR-54009	13,339	13,328	T	T	G	26%	Nsp10	N105K
NR-54974	14,679	14,668	T	T	C	22%	Nsp12	No AA Change
NR-54009	14,679	14,668	T	T	C	18%	Nsp12	No AA Change
NR-54974	22,114	22,103	T	T	C	12%	Spike	No AA Change
NR-54009	22,114	22,103	T	T	C	11%	Spike	No AA Change
NR-54974	25,806	25,786	A	A	G	6%	ORF3a	No AA Change
NR-54009	25,806	25,786	A	A	G	5%	ORF3a	No AA Change
NR-54974	28,237	28,217	G	G	T	6%	ORF8	R115L
NR-54009	28,237	28,217	G	G	T	90%	ORF8	R115L
NR-54974	28,368	28,348	G	G	A	90%	N	R32H
NR-54009	28,368	28,348	G	G	A	9%	N	R32H
NR-54974	29,821	29,801	T	T	G	92%	3′UTR	noncoding
NR-54009	29,821	29,801	T	T	G	12%	3′UTR	noncoding

**Table 5 viruses-13-02434-t005:** Twenty-five fixed variants present in BQ-RSA-p4 and BQ-RSA-p5 matching those in NR-54009 and NR-54974.

Position in MN908947 Wuhan-Hu-1 Sequence	Position in EPI_ISL_ 678615 Reference Sequence	Reported MN908947 Wuhan-Hu-1 Sequence	Reported EPI_ISL_ 678615 Reference Sequence	Nucleotide in NR-54009 and NR-54974	Variant Frequency BQ-RSA-p4	Variant Frequency BQ-RSA-P5	Gene	Amino Acid Mutation
174	172	G	T	T	95%	100%	5′ UTR	No AA Change
241	239	C	T	T	94%	100%	5′ UTR	No AA Change
1059	1057	C	T	T	94%	100%	Nsp2	T85I
2692	2690	A	T	T	90%	100%	Nsp2	No AA Change
3037	3035	C	T	T	94%	100%	Nsp3	No AA Change
5230	5228	G	T	T	74%	100%	Nsp3	K837N
10,323	10,321	A	G	G	95%	100%	Nsp5	K90R
11,287–11,295	11,285	GTCTGGTTTT	G/indel	G/indel	71%	100%	Nsp6	ΔSGF (aa106–108)
14,408	14,397	C	T	T	97%	100%	Nsp12	P323L
21,614	21,603	C	T	T	78%	100%	Spike	L18F
21,801	21,790	A	C	C	96%	100%	Spike	D80A
22,206	22,195	A	G	G	93%	100%	Spike	D215G
22,281–22,289	22,275	CTTGCTTAC	C/indel	C/indel	92%	100%	Spike	ΔLAL (aa242–244)
22,813	22,793	G	T	T	90%	100%	Spike	K417N
23,012	22,992	G	A	A	96%	100%	Spike	E484K
23,063	23,043	A	T	T	96%	100%	Spike	N501Y
23,403	23,383	A	G	G	96%	100%	Spike	D614G
23,664	23,644	C	T	T	90%	100%	Spike	A701V
25,563	25,543	G	T	T	95%	100%	ORF3a	Q57H
25,784	25,764	G	T	T	89%	100%	ORF3a	W131L
25,904	25,884	C	T	T	89%	100%	ORF3a	S171L
26,456	26,436	C	T	T	88%	100%	E gene	P71L
27,670	27,650	G	T	T	90%	100%	ORF7a	V93F
28,253	28,233	C	T	T	99%	92%	ORF8	No AA Change
28,887	28,867	C	T	T	98%	99%	N gene	T205I

**Table 6 viruses-13-02434-t006:** Variants in the BQ-RSA-p4 and BQ-RSA-p5 specific to virus grown on Calu-3 cells.

Position in MN908947 Wuhan-Hu-1 Sequence	Position in EPI_ISL_ 678615 Reference Sequence	Reported MN908947 Wuhan-Hu-1 Sequence	Reported EPI_ISL_ 678615 Reference Sequence	Nucleotide in NR-54974	Variant Frequency BQ-RSA-p4	Variant Frequency BQ-RSA-P5	Gene	Amino Acid Mutation
11,020	11,018	C	C	T	87%	99%	Nsp6	No AA Change
13,339	13,328	T	T	G	89%	95%	Nsp10	N105K
28,368	28,348	G	G	A	93%	99%	N gene	R32H
29,821	29,801	T	T	G	90%	98%	3′ UTR	noncoding

**Table 7 viruses-13-02434-t007:** De novo variants in the BQ-RSA-p4 and -p5 stocks.

Position in MN908947 Wuhan-Hu-1 Sequence	Position in EPI_ISL_ 678615 Reference Sequence	Reported MN908947 Wuhan-Hu-1 Sequence	Reported EPI_ISL_ 678615 Reference Sequence	Nucleotide Variant in BQ-RSA-p4 and -p5	Variant Frequency BQ-RSA-p4	Variant Frequency BQ-RSA-p5	Gene	Amino Acid Mutation
2969	2967	A	A	G	3%	16%	Nsp3	M902V
18,535	18,524	A	A	G	9%	56%	Nsp14	I6091V
21,633	21,622	T	T	C	1%	8%	S	L24S
23,044	23,024	A	A	G	4%	17%	S	No AA Change
24,619	24,599	A	A	G	4%	16%	S	No AA Change
26,250	26,230	C	C	T	16%	11%	E	No AA Change
26,453	26,533	T	T	G	8%	54%	E	V70G
26,465	26,545	T	T	C	3%	16%	E	L74P

**Table 8 viruses-13-02434-t008:** Variants present only in NR-54974-p5 at a frequency of greater than 5%.

Position in MN908947 Wuhan-Hu-1 Sequence	Position in EPI_ISL_ 678615 Reference Sequence	Reported MN908947 Wuhan-Hu-1 Sequence	Reported EPI_ISL_ 678615 Reference Sequence	Nucleotide Variant in NR-54974-p5	Variant Frequency	Gene	Amino Acid Mutation
9693	9691	C	C	T	13%	Nsp4	A3143V
25,406	25,386	T	T	G	12%	ORF3a	M5R
25,418	25,398	C	C	T	5%	ORF3a	Y9I
26,250	26,230	C	C	T	45%	E	No AA Change
26,461	26,442	CTTCTG	CTTCTG	C	13%	E	5nt deletion
29,274	29,254	C	C	T	9%	N	T334I
29,659	29,639	C	C	T	35%	ORF10	No AA Change

## Data Availability

Appendix A contains accession numbers for sequence data generated with samples from BIOQUAL. BEI sequence data is available upon request.

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
