# Peer review of "Propagation of SARS-CoV-2 in Calu-3 Cells to Eliminate Mutations in the Furin Cleavage Site of Spike"

_viruses, 2021, doi:10.3390/v13122434_

Round 1

Reviewer 1 Report

In the present manuscript, the authors indagate how the furin cleavage site in conserved according to the propagation method. In particular, they test such hypothesis in a cell line-dependent manner. As proof of concept, they employ an in vivo hamster model to recapitulate their findings.

The manuscript is really well-written. The introduction is inclusive of a complete state-of-art, results are presented in a logical and consequential manner, methods are informative to detail, yet not excessive. Conclusions are widely supported by the results, which are were structured and straight to the point.

Strength: the information contained in here are well documented and crucial for the research on SARS-CoV-2 worldwide. Every laboratory working in the field should be aware of these findings. Although it lacks biological novelty (see here below), it gives a methodological point of view of paramount importance, which could potentially impact thousands of laboratories.

Weakness: the novelty is widely abrogated by a previous paper (n. 15 in the manuscript: Loss of furin cleavage site attenuates. B.A. Johnson et al. January 2021. Nature. https://doi.org/10.1038/s41586-021-03237-4), were authors indagate the biological significance of the deletion of the furin cleavage site, both in vitro and in vivo. The permissiveness of Vero cells towards the deleted pseudovirions was already reported here, together with the information about the need to conserve the furin cleavage site for infection in CaLu3 and hamsters. However, it was never demonstrated before that the furin cleavage site could be “rescued” by culture in CaLu3. I believe this is more of an editorial choice, rather than a reviewer’s one (given the good quality of the manuscript).

SARS-CoV-2 pathogenesis

Minor concern:

There are some minor typos throughout the text (es. Line 124 “Vero-hSLAM which are Cercopithecus aethiops kidney”, where “which are” is written in italic).

The number of animals employed is really low and it lacks of the virus grown in Vero as a control. However, I believe it proves the point of the manuscript as a proof of principle nonetheless, based on the above-mentioned published manuscript.

The quality of the images is low, please consider uploading a higher resolution ones.

Reviewer 2 Report

Review of Viruses-1490103

Title: Propagation of SARS-CoV-2 in Calu-3 cells to eliminate mutations in the furin cleavage site of spike

Summary: This study investigated that propagation of SARS-CoV-2 on Calu-3 cells eliminates the accumulation of mutations in the furin cleavage site. Notably, there are alternative variants that accumulate at the same nucleotide positions in virus populations grown on Calu-3 cells at multiple independent sites. Furthermore, the authors found that a Calu-3 derived SARS-CoV-2 virus stock remained pathogenic to hamsters harboring the Calu-3 specific genetic variants.

Issues that should be resolved:

  • I may accept that Calu-3 can work as a better substrate for SARS-CoV-2 propagation than Vero derivatives, based on the results provided. In lines 437-444, however, the authors mentioned that ‘as long as SARS-CoV-2 stocks are being propagated in cells from different clinical isolates, our study supports a workflow where deep sequencing of a stock using analytical tools that can detect low frequency mutations needs to occur prior to animal challenge, and a control group infected with this virus is needed to assess the impact of any cell culture adapted mutations. Understanding virus diversity beyond the level of the consensus sequence is essential to identify underlying mutations that may alter the virus pathogenicity and we propose that it should be required for every in vivo challenge study.’ Then, how can we say that “the mutations identified are true genetic changes that can be called for each residue?” What if the mutations are decreasing and gone at last in the next passage step? What if viral characteristics determined in vitro or in vivo are the sum of functional balance orchestrated by all the genetic variants? Then, which is the main determinant? Major variant or minor variant? What if the mutations provided in the results were called marginally based on the proportion of genetic variations?

  • In an animal experiment, the authors infected hamsters with 1:10 dilution of the p4 stock. What is a viral titer of the 1:10 diluted inoculum? How many viral copies are present in the 1:10 diluted inoculum? Are viral copies determined at 2 days post-challenge replicated titers? Or, just remained in the oral cavity and reduced titers without being replicated? Did you check out how infectious SARS-CoV-2 particles might be present in the oral swab samples collected at 2, 4, or 7 days post-challenge? Without the viral titer results of the 1:10 diluted inoculum, it cannot be said that SARS-CoV-2 truly replicated in the respiratory organs of the hamsters. That means the viral genetic variations provided in the animal experiment should be confirmed again. So, I think the authors should clarify the main points of this study, as presented in the title, and revise the manuscript by removing the over-interpretation parts of genetic variations in the animal study.
